# Population genomics and haplotype analysis in spelt and bread wheat identifies a gene regulating glume color

Michael Abrouk [1,7], Naveenkumar Athiyannan [1,7], Thomas Müller[2,7], Yveline Pailles[1], Christoph Stritt[2], Anne C. Roulin[2], Chenggen Chu[3], Shuyu Liu[3], Takumi Morita[4], Hirokazu Handa[5], Jesse Poland [6], Beat Keller [2] & Simon G. Krattinger [1✉]

The cloning of agriculturally important genes is often complicated by haplotype variation across crop cultivars. Access to pan-genome information greatly facilitates the assessment of structural variations and rapid candidate gene identification. Here, we identified the *red glume 1* (*Rg-B1*) gene using association genetics and haplotype analyses in ten reference grade wheat genomes. Glume color is an important trait to characterize wheat cultivars. Red glumes are frequent among Central European spelt, a dominant wheat subspecies in Europe before the 20th century. We used genotyping-by-sequencing to characterize a global diversity panel of 267 spelt accessions, which provided evidence for two independent introductions of spelt into Europe. A single region at the *Rg-B1* locus on chromosome 1BS was associated with glume color in the diversity panel. Haplotype comparisons across ten high-quality wheat genomes revealed a *MYB* transcription factor as candidate gene. We found extensive haplotype variation across the ten cultivars, with a particular group of *MYB* alleles that was conserved in red glume wheat cultivars. Genetic mapping and transient infiltration experiments allowed us to validate this particular *MYB* transcription factor variants. Our study demonstrates the value of multiple high-quality genomes to rapidly resolve copy number and haplotype variations in regions controlling agriculturally important traits.

[1] Center for Desert Agriculture, Biological and Environmental Science and Engineering Division (BESE), King Abdullah University of Science and Technology (KAUST), Thuwal, Saudi Arabia. [2] Department of Plant and Microbial Biology, University of Zurich, Zollikerstrasse 107, Zurich, Switzerland. [3] Texas A&M AgriLife Research, Amarillo, TX, USA. [4] Department of Agricultural and Life Sciences, Kyoto Prefectural University, Kyoto, Japan. [5] Laboratory of Plant Breeding, Graduate School of Life and Environmental Sciences, Kyoto Prefectural University, Kyoto, Japan. [6] Department of Plant Pathology, Kansas State University, Manhattan, KS, USA. [7] These authors contributed equally: Michael Abrouk, Naveenkumar Athiyannan, Thomas Müller. ✉email: simon.krattinger@kaust.edu.sa

The recent completion of high-quality pan-genome and high-throughput genotyping projects provides the basis for a species-wide understanding of genome variations, which also facilitates the cloning of agriculturally important genes[1–5]. Wheat is one of the most important cereal crops that serves as a staple food for more than two billion people[6]. The two most widely cultivated wheat species today are tetraploid durum wheat, also called pasta wheat (*Triticum turgidum* ssp. *durum*) and hexaploid bread wheat (*T. aestivum* ssp. *aestivum*). Tetraploid and hexaploid wheats have exceptionally large and highly repetitive genomes. Because of this complexity, wheat genomics was lagging behind other major crop species. For example, chromosome-scale reference assemblies of the 10.5 Gb durum cultivar Svevo[7] and the 16 Gb bread wheat cultivar Chinese Spring[6] only became available during the past 2 years. These reference genomes provided a first detailed insight into the structure and organization of the complex wheat genomes, but they did not allow to assess species-wide genomic variations. To address the question of global variation in wheat genomes, ten chromosome-scale wheat assemblies, mostly representing elite cultivars, were recently completed in the frame of the "10+ Wheat Genomes Project"[8].

Spelt (*T. aestivum* ssp. *spelta*) is a subspecies of hexaploid wheat that is mainly cultivated in Central Europe and northern Spain as a high-value niche product. Spelt, however, was a staple crop in Europe from the Bronze Age until about the beginning of the 20th century[9]. In some regions of Central Europe, particularly in southern Germany and Switzerland, spelt even replaced tetraploid emmer as the dominant wheat species in the early Iron Age (around 750 BC)[10]. In 1930, spelt still accounted for ~40% of the Central European wheat production area[11]. Despite being a domesticated wheat, spelt shows some morphological characteristics that resemble non-domesticated grass species, including a brittle rachis and grains that are tightly surrounded by tenacious glumes (Fig. 1). Although these characteristics are beneficial for the dispersal and protection of seeds in wild plants, they are undesirable for mechanical harvesting and processing, which is one of the main reasons for the replacement of spelt with free-threshing bread wheat in the 20th century. The two hexaploid

wheat subspecies *aestivum* and *spelta* can be freely intercrossed, which breeders continue to exploit to transfer agronomically important genes from spelt into the bread wheat gene pool[12,13].

Hexaploid wheat emerged after two independent hybridization events that involved three diploid wild grass species of the Triticeae tribe. The first hybridization between wild einkorn (*T. urartu*, AA genome) and a close relative of *Aegilops speltoides* (BB genome) formed tetraploid wild emmer (*T. turgidum* ssp. *dicoccoides*, AABB genomes). Bread wheat arose after a second hybridization of domesticated emmer (*T. turgidum* ssp. *dicoccum*, AABB genomes) with the D genome of wild goatgrass (*Ae. tauschii*). The first hybridization occurred several hundred thousand years ago, whereas the second hybridization is thought to have occurred recently in a field of domesticated emmer ~10,000 years ago[14]. Despite the close relatedness of spelt and bread wheat, the population structure and agricultural history of spelt is still ambiguous and has not been well studied on a whole-genome level. It has been suggested that bread wheat and spelt have a common, monophyletic origin in the Fertile Crescent. During its migration to Europe, a free-threshing hexaploid wheat might have hybridized with a hulled tetraploid emmer wheat, which gave rise to European spelt[15–17]. This evolutionary model of a tetraploid wheat introgression would have resulted in genetic differentiation of the A and B subgenomes between bread wheat and European spelt, whereas the D subgenome would show a high degree of similarity[18]. A prominent trait that is found in many Central European spelt cultivars is red glumes. The traditional cultivar names of many European spelts, for example, "Oberkulmer Rotkorn" and "Altgold Rotkorn", contain the term "red" ("rot" in German) that refers to the glume color of mature spikes. Glume color is also an important trait to distinguish/ characterize bread wheat cultivars. Genetic studies revealed that glume color in hexaploid bread wheat is controlled by the three loci *Rg-A1* (1AS), *Rg-B1* (1BS), and *Rg-D1* (1DS) on the homoeologous group 1 chromosomes. Three dominant alleles *Rg-A1b*, *Rg-B1b*, and *Rg-D1b* are responsible for the red glume color, respectively, of which *Rg-B1b* (also referred to as *Rg1*) is the most frequent in hexaploid wheat. In addition, a black glume allele (*Rg-A1c*) and a smokey-gray glume allele (*Rg-D1c*) have been identified[19–22]. Homoeoalleles conferring dark glumes have also been described in various diploid wild wheat relatives of the genus *Aegilops* and in diploid and tetraploid *Triticum* species, indicating that genes controlling glume color were already present in the common ancestor of Triticeae[23]. The biochemical nature of ear coloration remains unclear. Transcriptional analysis of flavonoid biosynthesis structural genes suggested that *Rg* genes are activators of the phlobaphene/3-deoxyanthocyanidin branch of flavonoid biosynthesis[24].

Here, we used genotyping-by-sequencing (GBS)[25,26] to genotype 267 spelt accessions collected from the entire range of spelt cultivation. This comprehensive analysis of the spelt gene pool allowed us to unravel population structure and agricultural history with genome-wide marker coverage. We further demonstrate the usefulness of high-throughput genotyping, association genetics, and haplotype analyses in the ten high-quality wheat genomes by isolating the *Rg-B1* gene.

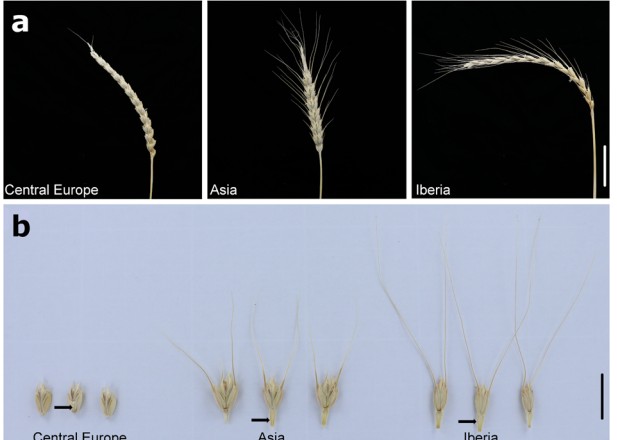

**Fig. 1 Spike morphology and spikelet disarticulation in spelt.**
**a** Representative spikes of spelt accessions collected in Central Europe, Asia, and the Iberian Peninsula. Scale bar = 5 cm. **b** Spikelet disarticulation of spelt. Spikelet disarticulation in plants with brittle rachis can be classified as barrel-type (upper rachis segment pressed against the lower spikelet) and wedge type (rachis segment pointing down). Central European spelt shows a barrel-type disarticulation, whereas disarticulation in Asian and Iberian spelt is of the wedge type. Scale bar = 2 cm. Arrows point to the rachis segment after spikelet disarticulation.

## Results

**Spelt population structure and demographic history.** Most of the European spelt accessions analyzed in this study were collected during the 1930s. Interestingly, many accessions from the Iberian Peninsula were morphologically more similar to Asian spelt than to Central European spelt accessions (Fig. 1). For example, spikes of many Spanish and Asian spelt accession carry awns (needle-like structures formed at the end of the floret),

whereas Central European spelt is generally awnless (Fig. 1a). Also, the spikelet disarticulation, which refers to the breakpoint of the rachis in relation to the spikelet, differed between Iberian and Central Europe spelt (Fig. 1b). We used GBS to assess the genetic variation of 267 spelt and 75 bread wheat accessions (Supplementary Data 1). The GBS returned 55,638 biallelic SNPs that passed the filtering criteria of missing rate ≤20%. Most variants (60%) were rare with a minor allele frequency (MAF) below 5% (Table 1, Supplementary Fig. 1). SNPs were distributed across the entire genome, with increased frequency in gene-rich, telomeric regions (Supplementary Fig. 2). Of the 55,638 variants, 19,379, 22,803, and 12,023 located on the A, B, and D subgenomes, respectively, whereas 1433 variants were unanchored. The fraction of variants in or close to genes was 37%; 7465 were gene proximal (±2 kb of coding sequence (CDS)), 8677 in exons, and 4293 in introns.

Principal component analysis (PCA), phylogeny, and admixture analyses revealed a separation of spelt into three gene pools comprising accessions collected in Asia, Central Europe, and the Iberian Peninsula (Fig. 2). In the PCA, Asian spelt accessions clustered together with bread wheat, whereas Central European and the Iberian spelts formed two distinct groups. The first principal component mainly separated European spelt from bread wheat and Asian spelt. Although the bread wheat accessions used in this study were collected in Central Europe, they were genetically closer to Asian than to the Central European spelt. The second axis separated Iberian spelt from the other two gene pools (Fig. 2a). Accessions that originated from artificial crosses between Central European spelt and bread wheat located in the middle of the first axis, confirming that the identified variants are correctly differentiating the populations. Two spelt accessions collected in America also localized within the wheat-spelt crosses, indicating a hybrid origin. Four accessions collected in Africa (one from Morocco and three from Ethiopia) clustered with the Central European spelt, possibly indicating that they were brought into Africa from Europe. Twenty-three spelt (8.7%) and one bread wheat accession showed discrepancies between their indicated geographic origin and the PCA cluster (Supplementary Data 1). The most likely reasons for this are erroneous passport information or mistakes during propagation of genebank material. Alternatively, this pattern might reflect interchange of germplasm between different regions before collection. A maximum-likelihood tree also provided evidence for three spelt gene pools (Fig. 2b). Inference of population structure was done using the ADMIXTURE tool, assuming various ancestral populations $K$, ranging from 2 to 10. The split into the three main gene pools revealed by PCA and phylogenetic analysis was apparent at $K = 3$ (Fig. 2c). The minimal cross-validation error value was obtained at $K = 7$ (Supplementary Fig. 3). Even with increasing $K$, the Iberian spelt remained a unique, homogenous population that did not split further. Compared to $K = 3$, the Central European spelt accessions were further divided into four different populations at $K = 7$, reflecting spring spelt, northern and southern accessions from Germany, and Switzerland (Fig. 2c).

Similar patterns as for the whole-genome analyses were observed for the three subgenomes (Supplementary Fig. 4). PCAs of the A and B subgenomes revealed a picture very similar to the whole-genome analysis (Supplementary Fig. 4a, b) with a clear separation into Central European, Asian, and Iberian populations. The situation was slightly different for the D subgenome (Supplementary Fig. 4c). Although a separation into the three populations was still observed, the distribution of the accessions was more compressed. The total variation explained by the two first principal components was lower for the D subgenome (9.3%) compared with the A (17.28%) and B (17.32%) subgenomes, respectively. These results indicate that the A and B subgenomes show a higher degree of divergence between the three spelt gene pools than the D subgenome, similar to what was previously reported for bread wheat[27]. These results are in agreement with a

**Table 1 Number of SNPs across different sample groups.**

| Samples | Number of SNPs | SNPs with MAF ≥ 1% | SNPs with MAF ≥ 5% |
|---|---|---|---|
| All (n = 339) | 55,638 | 35,642 | 23,554 |
| Wheat (n = 75) | 56,812 | 32,387 | 22,966 |
| Spelt/wheat crosses (n = 27) | 58,624 | 30,483 | 23,033 |
| Spelt (n = 237) | 57,327 | 34,402 | 23,207 |

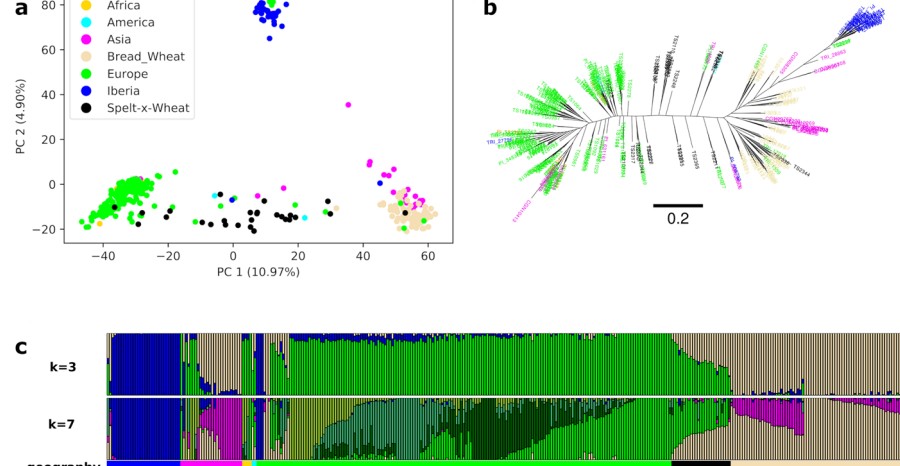

**Fig. 2 Whole-genome analysis of spelt and bread wheat. a** Principal component analysis (PCA) across the 339 spelt and bread wheat accessions. Samples are colored according to subspecies and geographical origin. **b** Maximum-likelihood tree constructed with SNPhylo. The colors used to label the accessions are identical to **a** and the branch size is indicated below the tree. **c** ADMIXTURE ancestry coefficients (*K* = 3 and *K* = 7) for the 339 accessions of spelt and bread wheat. Stacked bars represent accessions and colors represent ancestry components. Accessions are ordered according to subspecies and geographic origin.

common origin of the bread wheat and spelt D subgenome. We confirmed that the differences observed in the D subgenome analyses are not a consequence of the lower marker number per se. For this, we randomly sub-sampled 3500 subgenome-specific SNPs and repeated the PCA (Supplementary Fig. 5). The total variation explained by these sub-samples was similar to the PCAs constructed with all the markers. Also, we are not aware of any genomic variation in the D subgenome compared with the A and B subgenomes (TE content, gene number) that could explain the reduced diversity as a result of a technical artifact[6].

Considering the geographic proximity, it is surprising that Iberian and Central European spelts show such a clear morphological (Fig. 1) and genetic (Fig. 2) differentiation. As a first step toward understanding the possible origin of Iberian spelt, we used a coalescent-based evolutionary approach[28] to model and test three evolutionary scenarios: (i) a single introduction of Iberian and Central European spelt into Europe following a common westward migration from the Fertile Crescent, (ii) independent origins of Iberian and Central European spelt with independent introductions into Europe, and (iii) a hybrid model that assumed a common origin but allowed for a recent episode of gene flow from Asian to Iberian spelt. In all these models, the ancestral population was either assumed to be the Asian spelt population or an unsampled "ghost" population, resulting in a total of six models (Supplementary Fig. 6). Only D subgenome variants were considered because of the recurring gene flow from wild and cultivated tetraploid wheat (AABB genomes) into the cultivated hexaploid gene pool[27]. The model with the highest likelihood suggested an independent origin of Iberian and Central European spelt (AIC = 15,536.17), indicating that these two populations might have independent migration histories (Supplementary Table 1, Supplementary Table 2). Divergence times of Iberian and Central European spelt from their respective ancestral populations were estimated at 1218 (CI = 1018–1280) and 1023 (CI = 1014–1072) generations ago, respectively.

**Dissecting haplotype variation at the red glume Rg-B1 locus.** Red glume color is a characteristic trait that is frequently found in Central European spelts, as well as in many bread wheat cultivars (Fig. 3a). We performed a genome-wide association study based on the glume color of 102 Central European spelt accessions (66 white and 36 red, Supplementary Data 1). The glume phenotypes were extracted from historic data deposited in the Swiss National genebank (https://www.bdn.ch/). The genome-wide association study (GWAS) revealed a single peak on the short arm of chromosome 1B (Fig. 3b, Supplementary Fig. 7), corresponding to a confidence interval of ~2 Mb that spanned physical positions 2.24–4.17 Mb in the Chinese Spring RefSeq v1.0[6]. This physical position coincides with the Rg-B1 locus[21,22]. The associated region contained 33 annotated high-confidence genes in Chinese Spring (Supplementary Table 3), of which a single copy of a predicted R2R3-MYB-like transcription factor (TraesCS1B02G005200) was identified as the most promising candidate gene. MYB transcription factors have been identified as major determinants of grain, kernel, fruit, and flower pigmentation in plants by regulating the biosynthesis of flavonoids[29–34]. In the white glume wheat cultivar Chinese Spring[19], a 8.1 kb Mutator-like transposable element (MULE) inserted in the second intron of TraesCS1B02G005200 (Fig. 3c).

To assess the allelic variation across various wheat cultivars, we performed a haplotype analysis in the Rg-B1 interval across ten high-quality wheat genomes[8], which revealed extensive allele and copy number variation of the MYB transcription factor gene. In total, we identified 26 paralogs across the ten genomes, ranging

from zero in the Swiss bread wheat line ArinaLrFor to seven in spelt accession PI 190962. These 26 paralogs all located within the first 10 Mb of the short arm of chromosome 1B. The extensive structural variation represented 13 alleles that showed >93% sequence identity to TraesCS1B02G005200 (Fig. 3c, Supplementary Fig. 8). We classified the 13 alleles into five groups based on their gene structure and sequence similarities (Fig. 3c, Supplementary Fig. 8, Supplementary Table 4, Supplementary Table 5). Group 1, which includes the Chinese Spring reference MYB gene (TraesCS1B02G005200), contained six alleles (rg-B1a_h1–rg-B1a_h6). Characteristic for the group 1 alleles is that they all carry the same Mutator-like element in intron 2. Group 2 contained one allele (rg-B1a_h7) that was found in Jagger, Norin 61, and PI 190962. rg-B1a_h7 had the transposon insertion in intron 2 and an additional premature stop codon in the last exon, resulting in a predicted protein that lacked 66 amino acids at the C-terminus compared with rg-B1a_h1. The group 1 and 2 alleles were excluded as candidates for the red glume phenotype because rg-B1a_h7 is most likely a pseudogene and the group 1 alleles are the only alleles found in the white glume cultivars Chinese Spring (rg-B1a_h1)[19], CDC Stanley (rg-B1a_h1 and rg-B1a_h2), and CDC Landmark (rg-B1a_h3) (Pierre Hucl, personal communication), respectively. Alleles falling into groups 3 and 4 did not carry a transposon insertion and were mainly grouped based on sequence homology (Supplementary Table 4, Supplementary Fig. 8). The group 3 allele Rg-B1b_h1 was present both in Jagger and Norin 61 that have red/brown glumes[35,36]. As the other MYB alleles found in Jagger and Norin 61 belong to groups 1 and 2 (Fig. 3c, Supplementary Table 4), we hypothesize that Rg-B1b_h1 is a functional allele conferring the red glume phenotype. Group 4 contained two alleles, rg-B1a_h8 found in PI 190962 and rg-B1a_h9 found in the white glume Australian wheat cultivars Mace[37] and LongReach Lancer[38]. This indicates that group 4 alleles do not confer the red glume phenotype. Group 5 is represented by a single allele (rg-B1a_h10) that was only found in the assembly of spelt accession PI 190962. rg-B1a_h10 showed the greatest sequence divergence from the Chinese Spring rg-B1a_h1 allele (93.1% CDS identity; Supplementary Table 4, Supplementary Fig. 8) and carried a non-LTR short interspersed nuclear element type retrotransposon in intron 2. Allele rg-B1a_h10 was absent from the red glume cultivars Jagger and Norin 61[35,36]. In summary, the group 3 allele Rg-B1b_h1 emerged as the favored candidate gene because it was present both in Jagger and Norin 61.

To validate the candidate alleles, we developed group-specific markers based on specific nucleotide polymorphisms. In particular, we developed a co-dominant PCR-based marker for Rg-B1b_h1 based on a unique 47 bp InDel in the first intron that distinguished the group 3 alleles from all other groups. This marker produces a 334 bp amplicon specific for the group 3 alleles and a 381 bp amplicon for all other alleles (Fig. 3d). The markers were tested on a set of 53 Central European spelt accessions of our diversity panel. Consistent with the haplotype analysis in the ten high-quality wheat genomes, the group 1, 2, and 5 specific markers did not correlate with glume color in the spelt diversity panel (Supplementary Data 2). On the other hand, the markers specific for groups 3 and 4 amplified in all red glume spelt accessions, but not in the white glume accessions (Fig. 3d, Supplementary Data 2). These results show that most Central European spelt accessions carry multiple Rg-B1 alleles, but that only group 3 and 4 alleles were associated with red glumes. We further tested the group 3- and 4-specific markers on a collection of 96 bread wheat accessions of the Japanese wheat varieties core collection (JWC)[39] (Supplementary Data 3). Similar to the results in spelt, we observed an almost perfect correlation between the group 3-specific marker and glume color. The only exception was

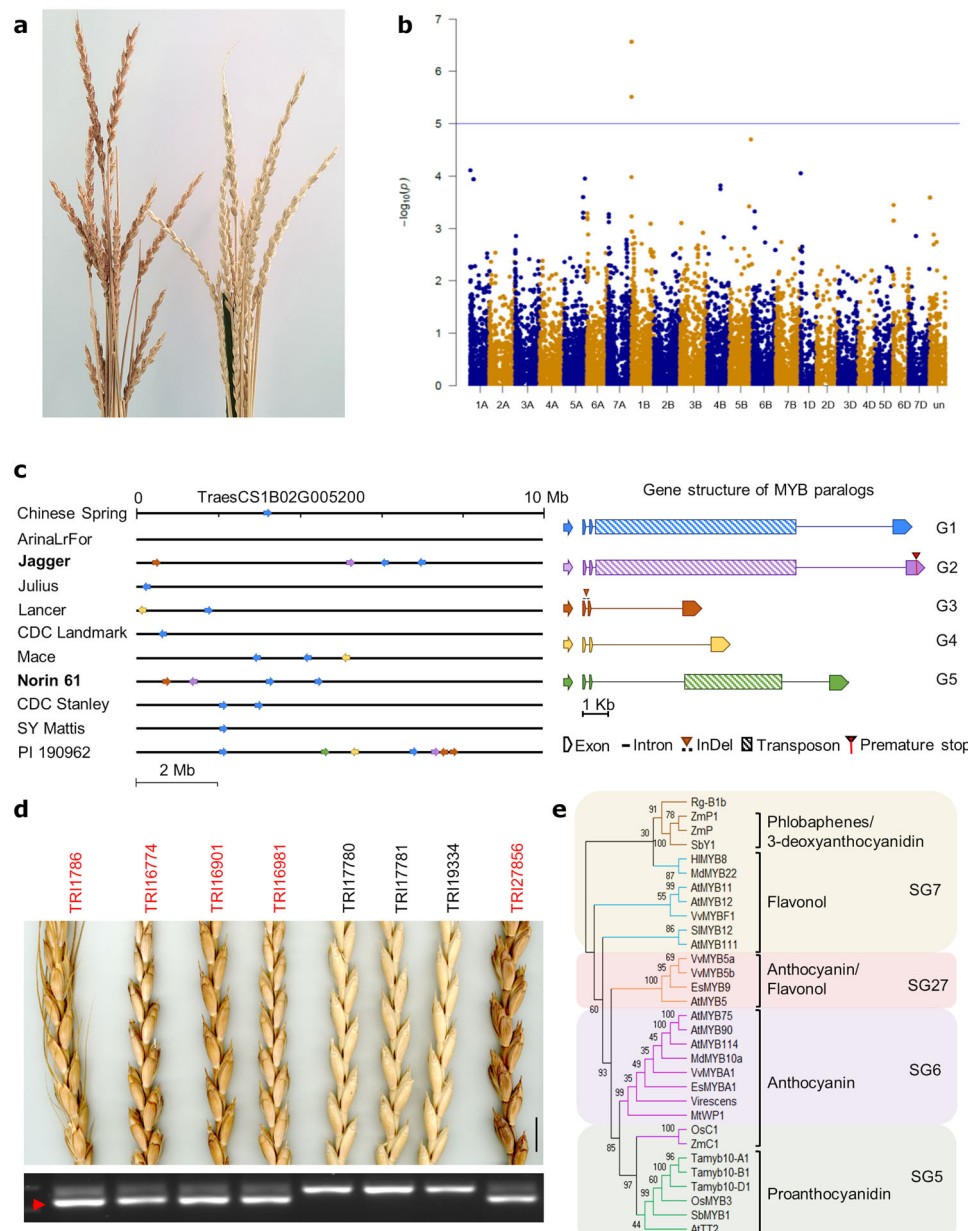

**Fig. 3 Haplotype variation at the red glume Rg-B1 locus. a** Images of a white glume spelt accession (right) and a red glume spelt accession (left).
**b** Manhattan plot showing a significant association for glume color at the Rg-B1 locus on chromosome arm 1BS. The physical confidence interval spans positions 2.24–4.17 Mb in the RefSeq v1.0 assembly of Chinese Spring. The region contains a MYB transcription factor gene (TraesCS1B02G005200).
**c** Copy number and allele variation for the candidate MYB transcription factor gene in ten high-quality wheat assemblies. Shown are the first 10 megabases of chromosome arm 1BS in the ten different wheat assemblies (left). Arrows indicate MYB-like transcription factor genes that are paralogous to TraesCS1B02G005200. Colors refer to one of five different allele groups (G1–G5) shown on the right. Group 3 is associated with red glumes in bread wheat and spelt. **d** Haplotype-specific PCR marker for the group 3 alleles. The red arrow points to the 334 bp amplicon specific for the group 3 Rg-B1 alleles. Scale bar = 1 cm. **e** Phylogenetic analysis of MYB transcription factors regulating the flavonoid biosynthesis pathway. SG = subgroup based on conserved amino-acid motifs[41]. The specific flavonoids that are linked to each subgroup are indicated.

accession JWC44 that amplified the 334 bp amplicon specific for the presence of group 3 alleles, but had pale spikes. We sequenced the entire Rg-B1b-CDS from JWC44 and found no polymorphism compared with Rg-B1b_h1, indicating that the gene is intact. The pale glume color in JWC44 might thus be caused by a second-site mutation outside the Rg-B1 locus or by mutations in the Rg-B1b_h1 regulatory regions. The group 4-specific marker amplified in all 46 red glume wheat accessions, but also in nine white glume accessions (including JWC44). The results confirm that the group 3 alleles are the most likely determinants of red glume color in

bread wheat and spelt. Since the group 3 alleles were never found alone, we cannot exclude the possibility that other MYB variants contribute to the red glume phenotype in the presence of a group 3 allele. In particular, the presence of a group 4 allele in all red glume accessions of the Central European spelt collection (Supplementary Data 2) and the Japanese wheat diversity panel (Supplementary Data 3) was intriguing, although wheat cultivars that had the group 4 allele in the absence of a group 3 allele were white. This prompted us to carefully re-evaluate the ten high-quality bread wheat sequences. Although the red glume cultivars

Jagger and Norin 61 did not carry a group 4 allele on chromosome 1B, we found a group 4 allele with 100% CDS identity to rg-B1a_h9 on chromosome unanchored and chromosome 1D of Jagger and Norin 61, respectively. rg-B1a_h9 in Norin 61 was located on a ~1 Mb segment of chromosome 1D that showed high sequence identity to the rg-B1a_h9 containing segment on chromosome 1B of Mace (Supplementary Fig. 9). Whether the 1D location of this segment in Norin 61 represents an assembly artifact or a true translocation could not be determined. The white glume wheat cultivar CDC Landmark also carried a rg-B1a_h9 allele on chromosome 1D. These results confirm that group 4 alleles are not determining glume color, but might contribute to the phenotype in the presence of a group 3 allele.

In addition to the spelt and bread wheat diversity panels, we observed co-segregation between the group 3 marker and glume color in 93 recombinant inbred lines of a bi-parental mapping population generated from a cross between the bread wheat lines TAM112 (red) and Duster (white) (Supplementary Data 4). Sequencing of 97 cDNA clones from red glume spelt accessions representing all haplotype groups revealed that the group 3 and 4 alleles were expressed, whereas expression of the group 1, 2, and 5 alleles was not detectable in red and white glume accessions. The genetic mapping thus provides strong evidence that group 3 alleles determine glume color in spelt and bread wheat.

**Characterization of the Rg-B1 group 3 alleles.** Both Jagger and Norin 61 carried the same Rg-B1 group 3 allele (Rg-B1b_h1; TraesJAG1B01G000800 and TraesNOR1B01G001100). To confirm the predicted gene model, we sequenced the cDNA of Rg-B1b_h1 from several red glume spelt accession and found an open-reading frame of 975 bp, translating into a predicted protein of 324 amino acids. This is 24 bp shorter than the predicted CDSs in Jagger and Norin 61. Gene models in the ten wheat genomes were predicted based on gene projections of the Chines Spring gene models[8]. Chinese Spring only carries a group 1 allele of the MYB transcription factor, which might have resulted in a wrong prediction of the gene models in Jagger and Norin 61. The spelt accession PI 190962 had two copies belonging to the group 3 alleles. The two alleles found in PI 190962 differed by one (Rg-B1b_h2) and four (Rg-B1b_h3) non-synonymous SNPs from Rg-B1b_h1, respectively (Supplementary Fig. 10). Interestingly, spikes of PI 190962 appear to be white (https://npgsweb.ars-grin.gov/gringlobal/ImgDisplay?taxid=406903; Assaf Distelfeld, personal communication). This could indicate that Rg-B1b_h2 and Rg-B1b_h3 are non-functional or that PI 190962 carries an unlinked suppressor gene. To test this, we developed allele-specific PCR markers based on the SNPs between Rg-B1b_h1, Rg-B1b_h2, and Rg-B1b_h3. The markers were tested on 21 red glume Central European spelt accessions of our collection (Supplementary Data 2). Four of these accessions amplified the Rg-B1b_h1-specific marker only. Thirteen accessions had the Rg-B1b_h1 and Rg-B1b_h3 alleles and four accessions had the Rg-B1b_h2 and Rg-B1b_h3 allele combination found in PI 190962. These results suggest that many spelt accessions carry more than one group 3 allele and that the white phenotype in PI 190962 is most likely caused by a second-site mutation. An unlinked suppressor of glume color in wheat has been described on chromosome 3A[23,40].

Rg-B1 encodes a putative R2R3-MYB-like transcription factor, the largest MYB transcription factor family in plants. R2R3-MYB transcription factors have a highly conserved N-terminal DNA-binding domain and a more variable modulator region at the C-terminus[41]. The closest homolog of Rg-B1b_h1 in maize[42] is ZmP1, a R2R3-MYB transcription factor regulating pericarp pigmentation in maize kernels[33]. The closest Arabidopsis homolog is AtMYB12, which regulates flavonoid accumulation[43]. R2R3-MYB transcription factors are classified into 28 subgroups based on conserved amino-acid motifs in their most C-terminal MYB domain[41]. Phylogenetic analyses indicate that Rg-B1b_h1 belongs to subgroup 7 (Fig. 3e) that has been associated with the biosynthesis of flavonols and phlobaphenes[41].

**Group 3 Rg-B1 alleles upregulate flavonoid biosynthesis genes.** To analyze whether Rg-B1 might regulate flavonoid biosynthetic genes in wheat, we performed RT-qPCR in 20 (eight red and 12 white glume) spelt accessions using wheat flavonoid biosynthesis pathway genes[44,45]. Transcript levels of the early flavonoid biosynthetic genes chalcone synthase (TaCHS), chalcone isomerase (TaCHI), and flavonoid 3'-hydroxylase (TaF3'H) were higher in red glume accessions compared with white glume accessions (Supplementary Fig. 11). These three genes are involved in the generation of common precursors of the flavonoid biosynthesis pathways[46]. In addition, transcript levels of flavone synthase (TaFNS) were also found to be upregulated in red glume accessions, which could indicate that the red pigment might be owing to the production of flavones. In sorghum, for example, the expression of F3'H and FNSII was associated with the production of the flavones apigenin and luteolin, resulting in light or dark brown pigmentation in injured leaves[47]. The transcript levels of the flavonoid biosynthesis genes flavanone 3-hydroxylase (TaF3H), dihydroflavonol 4-reductase (TaDFR), flavonol synthase (TaFLS), and anthocyanidin synthase (TaANS) were unaffected by glume color (Supplementary Fig. 11). Staining with diphenylboric acid 2-amino-ethyl ester (DPBA), a chemical that results in fluorescent compounds after specifically binding to flavonoids[48,49], produced higher fluorescence in red glume spelt accessions compared with white glume accessions (Supplementary Fig. 12a–c).

**Transient expression of Rg-B1 in Nicotiana benthamiana.** Overexpression constructs of Rg-B1 alleles Rg-B1b_h1 and Rg-B1b_h3 (35 S:Rg-B1b_h1:GFP and 35 S:Rg-B1b_h3:GFP) were transiently expressed in N. benthamiana. RT-qPCR revealed an upregulation of the flavonoid biosynthesis genes NbCHS, NbCHI, NbF3H, NbF3'H, and NbFLS in leaves expressing the two Rg-B1 alleles compared with the control (Fig. 4a). Similar to the observations made in spelt spikes (Supplementary Fig. 11), Rg-B1 did not affect DFR transcript levels in Nicotiana. DFR regulates the anthocyanidin pathway from dihydroflavonols. The lack of DFR expression in both wheat and N. benthamiana might indicate that neither proanthocyanidin nor anthocyanin are involved in the wheat red glume phenotype. Histochemical staining of N. benthamiana leaves with DPBA produced a fluorescent peak between 550 and 650 nm that was absent in the controls and in unstained leaves (Fig. 4b). This broad peak was previously reported to be associated with a "yellow-gold" fluorescence from the flavonol quercetin[48]. This could indicate that the pigments produced in spelt spikes (possibly flavones) and infiltrated Nicotiana leaves (possibly flavonols) might be different. Infiltration of a group 4 (rg-B1a_h9) and a group 5 (rg-B1a_h10) allele also resulted in a fluorescent peak, whereas the group 1 allele rg-B1a_h2 did not (Supplementary Fig. 12d). These results indicate that group 4 and 5 alleles also have the potential to induce flavonoid production when overexpressed in an artificial system. The group 1 alleles rg-B1a_h2, rg-B1a_h3, and rg-B1a_h4 carry a deletion of a conserved amino acid in the N-terminal R2 MYB domain, which might explain why they were not functional in Nicotiana.

In summary, the haplotype analysis, association and genetic mapping, and infiltration experiments demonstrate that the

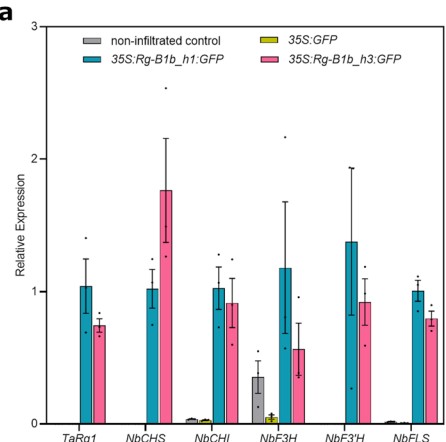 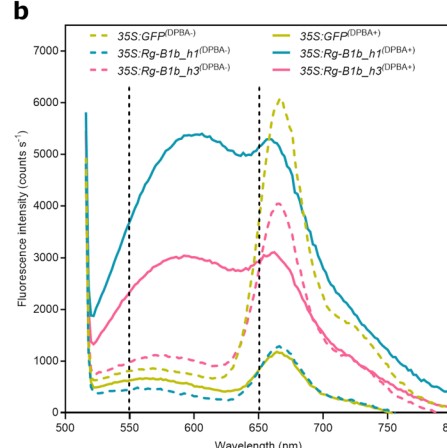

**Fig. 4 Transient expression of *Rg-B1* in *Nicotiana benthamiana*. a** Transcript levels of endogenous flavonoid biosynthetic genes in *N. benthamiana* infiltrated with the 35 S:GFP vector control and the *Rg-B1*-overexpressing constructs 35 S:*Rg-B1b_h1:GFP* and 35 S:*Rg-B1b_h3:GFP*. Error bars represent standard errors of three biological replicates. **b** Spectrofluorometric profile of agroinfiltrated leaves pre and post DPBA staining. The region between the two vertical dotted lines from 550 to 650 nm coincides with the previously reported peak for the flavonol quercetin[48]. The peak from 665 to 685 nm is autofluorescence from chlorophyll/chloroplasts[49].

glume color at the *Rg-B1* locus is controlled by a particular group of R2R3-MYB transcription factor variants.

## Discussion

The separation of hexaploid wheat into *aestivum* and *spelta* has mostly been based on morphological spike characteristics. Although spelt shows a narrow, elongated spike with brittle rachis and tenacious glumes, bread wheat has a more compact, square spike with a non-brittle rachis and free-threshing grains. It has been shown that the spike morphology in wheat is controlled by only a few loci. For example, the *Q* locus located on wheat chromosome arm 5AL represents a major determinant of spike morphology. *Q* encodes an AP2 family transcription factor that pleiotropically influences rachis fragility, glume shape and tenacity, and spike length[50]. The recessive *q* allele is found in European spelt and confers the typical spelt-like spike. The dominant *Q* allele of bread wheat shows higher transcript levels, resulting in more compact spikes with free-threshing grains. The *Q* allele differs from *q* by one non-synonymous polymorphism in the open-reading frame and five conserved polymorphisms in the promoter region[50]. Interestingly, certain Asian spelt accessions have been reported to carry the *Q* allele found in bread wheat[51], which led to the hypothesis that *Q* has a regulatory function and that the expression level of *Q*, and hence control of the spike morphology, depending on the genetic background. Another genetic determinant of spike characteristics in cereals is the locus containing the two tightly linked genes *Non-brittle rachis 1* (*btr1*) and *Non-brittle rachis 2* (*btr2*) identified on chromosomes 3 of barley and diploid einkorn[52,53]. The two genes code for short proteins with no domain homology, and mutations in either of the two genes convert a brittle rachis into a non-brittle one, which is associated with cell wall thickness in the spikelet disarticulation zone. Our whole-genome marker analysis did not result in a separation of a common spelt from the bread wheat gene pool. Asian spelt was indistinguishable from bread wheat based on our molecular analysis, whereas Central European and Iberian spelt formed two clearly distinct groups. Hence, although a separation into ssp. *spelta* and ssp. *aestivum* might be justified based on morphological characteristics, molecular evidence does not support this. Instead, we propose to split the hexaploid wheat gene pool into three groups, (*i*) bread wheat/Asian spelt, (*ii*) Central European spelt, and (*iii*) Iberian spelt. This is in agreement with

previous analyses that were performed with a limited number of molecular markers[15,17,18,54].

Hexaploid wheat evolved in the Fertile Crescent during the Neolithic Revolution. It is thus very likely that Asian spelt represents the most ancient of the three spelt gene pools. In contrast to Asian spelt, spelt accessions from Europe markedly differed from Asian spelt/bread wheat. Various reports suggested that Central European spelt emerged from hybridization of free-threshing hexaploid wheat with a hulled tetraploid emmer[15–18], a hypothesis that is supported by experimental crosses[55]. A DNA sequence from 8200 year-old charred wheat grain excavated at the Çatalhöyük archeological settlement in Central Turkey revealed a high similarity to modern Central European spelt[56]. This might indicate that a Central European-like spelt was already present ~6200 BC at a site that lies along the migration route from the Fertile Crescent to Europe. It would also indicate that the hybridization that gave rise to Central European spelt occurred somewhere between the Fertile Crescent and Central Turkey. A very interesting observation is that spelt accessions from the Iberian Peninsula formed a clearly distinct cluster, which raises the question about the origin of Iberian spelt. Our demographic modeling provides the first possible answer to this question and the best model suggested an independent introduction of Central European and Iberian spelt into Europe. The likelihood differences between the six models were comparable to other reports[57], supporting that our best performing model is plausible. Interestingly, a recent population genomics analysis found a genetic difference between ancient humans from Iberia and Central Europe[58], which proposed two independent migrations by Neolithic farmers with slightly different gene pools. The first migration route followed the Danube River into Central Europe and the second route went along the Mediterranean coast into the Iberian Peninsula. It is conceivable that these two ancient human populations also brought along and cultivated cereals of slightly different origins.

Because of its striking phenotype and importance for classifying bread wheat cultivars, the genetic inheritance of glume color was among the first traits that were systematically studied after the rediscovery of Mendel's laws in the early 1900s[24,59–61]. Almost 120 years later, we complement these pioneering studies with the identification of the DNA sequence that causes the red glume phenotype. In plants, pigments are the result of secondary

metabolites, including anthocyanin, proanthocyanidins, flavonols, flavones, and flavan-4-ols derivatives such as phlobaphenes and 3-deoxyanthocyanidins, which are synthesized through the phenylpropanoid and flavonoid pathways[41]. Genes involved in flavonoid biosynthesis can be divided into structural and regulatory genes. Three transcription factor families, MYB, basic-helix-loop-helix (bHLH), and WD-repeat (WDR), have been found to regulate flavonoid biosynthesis[41,46]. The disruption of flavonoid biosynthesis through transposon insertions in or close to regulatory R2R3-MYB transcription factors is a common mechanism[62]. For example, the presence/absence of a copia-like retrotransposon in exon 3 of a date palm R2R3-MYB transcription factor gene resulted in yellow and red date fruits, respectively[29]. The *Ruby2-Ruby1* MYB gene cluster regulates anthocyanin production and fruit pigmentation in Citrus and it has been found that transposon insertions have significant regulatory roles[63]. For example, the insertion of a copia-like retrotransposon upstream of *Ruby* results in cold-dependent accumulation of anthocyanins in blood oranges[64]. This study also highlights the effect of environmental conditions on pigment production. Besides temperature, several reports indicated that flavonoid biosynthesis is regulated by light[34,65]. We observed that light intensity had an effect on glume color in wheat (Supplementary Fig. 13), which is in agreement with previous observations indicating that wheat glume color is affected by environmental conditions[23,61].

Our study highlights the importance of pan-genome information for gene cloning in wheat. Chinese Spring only contained a single, non-functional *MYB* transcription factor allele at the *Rg-B1* locus and the extensive copy number and allelic variation at this locus would have remained hidden without access to the ten high-quality genomes. As demonstrated in this study, the completion of these ten high-quality genomes represents a step-change for gene cloning in wheat.

## Methods

**Plant material**. Grains of 75 *T. aestivum* ssp. *aestivum* and 267 *T. aestivum* ssp. *spelta* accessions were obtained from four germplasm collections (Supplementary Data 1). The spelt accessions were chosen to cover the entire cultivation range of the subspecies. Twenty-seven accessions were derived from recent crosses between bread wheat and spelt in breeding programs. The set included spelt from Central Europe, Southern Europe, Africa, Asia, and America. The *T. aestivum* ssp. *aestivum* accessions all originate from Central Europe (Supplementary Data 1).

**DNA extraction**. Single plants were grown and leaf segments of ~3 cm were sampled from the first leaf of 12-day-old seedlings. Frozen leaf tissue was ground with a Mixer Mill MM 400 (Retsch, Haan, Germany) for 60 sec at 13,000 rpm. Ground leaf tissue was suspended in 500 μl extraction buffer (1 M guanidine thiocyanate, 2 M sodium chloride, 30 mM NaAc, pH 6.0, 0.2% Tween 20) and incubated at 65 °C for 30 min. The lysate was centrifuged for 10 min at 2500 rpm and 300 μl of the supernatant was transferred onto DNA spin columns (Epoch Life Science, Inc, Texas, US). The columns were centrifuged at 13,000 rpm for 1 min, washed twice with 600 μl and 250 μl wash buffer (50 mM sodium chloride, 10 mM Tris/HCl, pH 8.0, 1 mM ethylenediaminetetraacetic acid (EDTA) in 70% ethanol) and centrifuged at 13,000 rpm for 1 min after each wash step. DNA was eluted from the columns with 100 μl TE buffer (10 mM Tris/HCl, pH 8.0, 1 mM EDTA).

**Genotyping-by-sequencing**. DNA samples were genotyped with GBS using *PstI* as a restriction enzyme by the Institute of Biotechnology, Cornell University, NY, USA. Samples were sequenced on six lanes of an Illumina HiSeq 2000 and six lanes of an Illumina NextSeq 500 yielding 1,915,955,758 reads in total.

**SNP calling**. Reads were mapped against the Chinese Spring IWGSC RefSeq v1.0[6] with bwa v0.7.15[66]. SNPs were called using TASSEL v5.2.31[67] with a read length of 64 bp. SNPs were filtered using vcftools v0.1.14[68] using the following criteria: (1) biallelic SNP, (2) SNPs with >20% missing data were discarded, (3) accessions with >20% missing information were discarded.

**Population genomic analyses**. PCA were performed using the python library scikit-learn v0.17.1[69]. Phylogenetic trees were computed using the SNPhylo software package[70] with the 54,205 anchored SNPs. Low-quality SNPs based on allele frequency (MAF ≤ 0.05) and missing data (≥20%) were removed using the SNPRelate package[71]. Multiple sequence alignments were done using MUSCLE[72] and phylogenetic trees were constructed by running DNAML from the PHYLIP package[73]. One hundred bootstraps using the phangorn package[74] were done and trees were visualized and annotated with FigTree v1.4.3. Structure analyses were performed with the ADMIXTURE software[75]. Data management and quality control operations were performed using vcftools v0.1.14[68] and PLINK v1.9[76]. We explored *K* values from 2 to 10 and determined the lowest cross-validated error rate.

**Demographic inference**. Demographic models were fitted to folded 2D site frequency spectra with fastSimcoal2 (version 2.6.0.3)[28]. Accessions displaying discrepancies between their indicated geographic origin and the PCA cluster were removed (Supplementary Data 1). Only D subgenome variants were considered because of the polyphyletic origin of the A and B subgenomes. We used biallelic SNPs filtered for missing data, and removed singletons. As fastSimcoal assumes sites to be neutral and unlinked, we removed genic SNPs and only retained SNPs at least 20 kb apart from each other, leaving a total of 2644 SNPs distributed among 209 spelt accessions from Asia, Central Europe, and Iberia.

Folded two-dimensional site frequency spectra were estimated with dadi[77]. The numbers of alleles were projected down from 36 to 24, from 324 to 254, and from 58 to 42 for the Asian, the Central European, and Iberian spelt, respectively, so as to maximize the number of segregating sites used to construct the SFS. A mutation rate of $1.3 \times 10^{-8}$ was assumed[78]. Further model specifications and parameter search ranges can be found in Supplementary Table 1. For each of the six models, 20 independent runs with 50 ECM cycles and 100,000 simulations per estimation step were performed. Confidence intervals were obtained with a parametric bootstrap approach in which 100 data sets were simulated with the parameter values of the best model, and parameter values then inferred from the simulated data.

**Genome-wide association study of glume coloration**. For the GWAS, 18,260 SNPs with a MAF > 0.05 from 102 spelt accessions were retained. The glume color was phenotyped based on photographs deposited in the national genebank of Agroscope, Switzerland (https://www.bdn.ch/). The association of SNPs with glume color was tested by a logistic regression model assuming additive genetic effects using PLINK software (v1.09)[76]. Two principal coordinates, obtained by multidimensional scaling, were included as covariates in the logistic regression model to control for population stratification. A quantile–quantile plot of GWAS was used to examine the *P* value distribution (Supplementary Fig. 7). A Manhattan plot was used to display the association *P* value for each SNP. Significant associations were identified above the threshold of *P* value = $1.0 \times 10^{-5}$. Both Q–Q plot and Manhattan plot were plotted using the "qqman" R package[79].

***MYB* sequence analyses in ten wheat genomes**. The CDS of the Chinese Spring *MYB* gene (TraesCS1B02G005200) was used for a BLASTbasic local alignment search tool comparison against the gene projections of the 10+ Wheat Genomes Project[8]. Hits with >90% identity on the total length of the query sequence were shortlisted and further analyzed. In order to retrieve unannotated *MYB* copies, a direct comparison with TraesCS1B02G005200 was performed against every chromosomes of each assembly using exonerate v2.2.0[80]. Alignments were done using the MUSCLE program in MEGA-X using default parameters[81]. Amino-acid sequences of Rg-B1 homologs from different species and various MYB transcription factors involved in plant pigmentation were retrieved from NCBI and were aligned using the MUSCLE program in MEGA-X using default parameters. A neighbor-joining tree was constructed with 1000 bootstrap replicates in MEGA-X[81].

**PCR conditions**. A 20 μl PCR reaction containing 100 ng of gDNA, 1× GoTaq Green Master Mix (M7122 Promega, USA), and 200 nM primers were used for *Rg-B1* marker screenings. Primer sequences are provided in Supplementary Data 5. A touch-down PCR protocol was run as follows: initial denaturation at 94 °C for 30 seconds; annealing at 62 °C for 30 seconds, decreasing by 0.5 °C/cycle; extension at 72 °C for 45 seconds, followed by repeating these steps for eight cycles. After enrichment, the program continued for 29 cycles as follows: 94 °C for 30 seconds, 58 °C for 30 seconds and 72 °C for 45 seconds. A 20 μl PCR reaction with High-Fidelity Phusion *Taq* polymerase (M0530, New England Biolabs) was performed to clone the full-length *Rg1* alleles following manufacturer's instructions.

**Cloning of *Rg-B1***. A 972 bp cDNA fragment without stop codon of the *Rg-B1b_h1* and *Rg-B1b_h3* alleles were amplified using primer pairs RgattBF2-RgattBR1 (Supplementary Data 5) and cloned into the entry vector pDONR207 using gateway BP clonase II (11789020, Thermo Fisher Scientific). *Rg-B1b_h1* and *Rg-B1b_h3* clones were confirmed by Sanger sequencing. CDSs without stop codon of *rg-B1a_h2*, *rg-B1a_h9*, and *rg-B1a_h10* alleles were synthesized and cloned into pDONR221 from GeneArt Gene Synthesis, Thermo Fisher Scientific. The constructs were then introduced into the binary vector pGWB5[82] to generate *35S:Rg-B1b_h1:GFP*, *35S:Rg-B1b_h3:GFP*, *35S:rg-B1a_h2:GFP*, *35S:rg-B1a_h9:GFP*, and *35S:rg-B1a_h10:GFP* using Gateway LR Clonase II (11791020, Thermo Fisher Scientific). The constructs in the pGWB5 backbone were transformed into *Agrobacterium tumefaciens* strain GV3101.

***A. tumefaciens* mediated transient expression of *Rg-B1* in *N. benthamiana*.** *A. tumefaciens* strains carrying the expression constructs (*35 S:Rg-B1b_h1:GFP; 35 S:Rg-B1b_h3:GFP, 35 S:rg-B1a_h2:GFP, 35 S:rg-B1a_h9:GFP, 35 S:rg-B1a_h10:GFP* and *35 S:GFP*) and the suppressor of post-transcriptional gene silencing p19 were grown in LB broth with appropriate antibiotics overnight at 28 °C at 200 rpm. *A. tumefaciens* cells were pelleted by centrifugation at 4000 rpm for 10 min, and resuspended in MMA induction buffer (10 mM MES, 10 mM $MgCl_2$, 100 μM acet-osyringone, pH 6.5). Cell density was measured at OD600 nm and adjusted to 0.8. After induction, *A. tumefaciens* cells were mixed with the p19-silencing-suppressor strain at a ratio of 1:1 prior to syringe infiltration into four-week-old *N. benthamiana* leaves. *A. tumefaciens* cells with pGWB5 vector expressing GFP (*35 S: GFP*) were used as a negative control. Leaf samples at 48–72 hours after agroin-filtration were collected for RT-qPCR.

**Quantitative real-time PCR.** A subset of 20 spelt accessions grown in a glasshouse illuminated with LX602C LED grow lights (Heliospectra, 20/4 hours day/night) were used for quantitative real-time PCR (RT-qPCR) studies. Spikelets from spikes 15 days after booting were collected in three biological replicates for RNA isolation. In the case of *N. benthamiana*, three leaf disks per *A. tumefaciens* infiltrated sample and three biological replicates per construct were collected for RNA isolation. RNA extraction was done using the Maxwell® RSC Plant RNA kit using a Maxwell® RSC 48 instrument (Promega). In all, 1 μg of RNA was used for the first-strand complementary DNA (cDNA) synthesis using the high-capacity cDNA reverse transcription kit following the user guidelines (Applied Biosystems catalog # 4368814). cDNA was further diluted 10 fold and 2 μl was used for RT-qPCR. RT-qPCR was performed for *Rg1* and flavonoid biosynthetic genes using primers listed in Supplementary Data 5. A 10 μl RT-qPCR reaction was set-up and run on an ABI QuntStudio 6 Flex Real-Time PCR machine using PowerUp SYBR green master mix (Applied Biosystems catalog # AS25741). The $2^{-\Delta\Delta CT}$ method was used to normalize and calibrate transcript values relative to endogenous controls *Ta.6863* (spelt)[83] and *tubulin (NtTub1)* (*N. benthamiana*).

**Flavonoid staining.** A diphenylboric acid 2-amino-ethyl ester (DPBA, also known as 2-aminoethyl diphenylborinate, Naturstoff reagent A) protocol was adapted for in situ staining of flavonoids[48]. In brief, *N. benthamiana* leaves harvested at 5 days after agroinfiltration were bleached in absolute ethanol. Bleached leaves were stained using freshly prepared 0.25% (w/v) DPBA and 0.00375% (v/v) Triton X-100 in absolute ethanol for at least 2 hours at room temperature. DPBA-stained leaves were homogenized in 1 ml of staining solution and centrifuged at 14,000 rpm for 10 minutes. A 200 μl aliquot in replicates was used to measure the fluorescence intensity in a spectrofluorometer excited at 488 nm, and emission spectra from 490 nm to 800 nm were recorded. In all, 100 mg of spelt outer glumes were ground with liquid nitrogen and stained with 2 ml of DPBA solution placed on a three-dimensional mixer for 24 hours. The extracts were centrifuged at 14,000 rpm for 10 minutes and replicates of fivefold diluted supernatants were used for fluorescence intensity measurement.

**Reporting summary.** Further information on research design is available in the Nature Research Reporting Summary linked to this article.

## Data availability

The raw sequence reads were deposited in the short read archive (SRA) on NCBI under the accession PRJNA498918. The VCF file and GWAS summary statistics are available on DRYAD repository under [10.5061/dryad.d7wm37pzs].

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

## Acknowledgements

We are grateful to the genebanks of the Leibniz Institute of Plant Genetics and Crop Plant Research (IPK), the Centre for Genetic Resources (CGN), the USDA National Plant Germplasm System, and the Swiss National Genebank for providing grains of the various accessions. We thank Simon Fluckiger and Helen Zbinden from the University of Zurich for technical assistance, Harold E. Bockelmann from USDA-ARS for valuable discussion on the USDA accessions, professor Pierre Hucl, and professor Curtis Pozniak from the University of Saskatchewan for providing information on the glume colors of CDC Stanley and CDC Landmark, professor Assaf Distelfeld from Tel Aviv University for commenting on the glume color of PI 190962, and professor Robert McIntosh from the University of Sydney for discussions on the designation of the Rg-B1 alleles.

## Author contributions

T.M., A.C.R., J.P., B.K., and S.G.K. conceived the study. M.A., C.S., and T.M. curated bioinformatic data. M.A., N.A, Y.P., C.S., and T.M. performed bioinformatics analyses. N.A., C.C., S.L., T. Morita, and H.H. performed molecular analyses and genetic mapping. B.K. and S.G.K. acquired funding. M.A., N.A., and S.G.K. wrote the original draft.

## Funding

This work was supported by the Swiss Federal Office for Agriculture (BLW) in the framework of NAP-PGREL (national plan of action for the conservation and sustainable utilization of plant genetic resources, project 05-NAP-O34), the University of Zurich and the University Research Priority Program "Evolution in Action", and the King Abdullah University of Science and Technology (KAUST).

## Competing interests

The authors declare no competing interests.
