## [Peer Review File · Communications Biology]

Reviewers' comments:

Reviewer #1 (Remarks to the Author):

Association of hexaploid wheat genes, responsible for agronomically important traits, with certain genomic sequences remains a challenge even after whole genome sequencing of wheat. The results obtained in the current study can be estimated as valuable, because for the first time since the early 1900s when geneticists began to study red glume color trait in wheat, the Rg gene has been fully described at the DNA level. In addition, in the reviewed paper, some new insights to the origin and history of hexaploid spelt wheat are reported. The article can be accepted after following revisions:

(1) Page 5 line 90-92: It is more correct to say "Genetic studies revealed that glume color in hexaploid wheat is controlled by three loci on the homoeologous group 1 chromosomes: Rg-A1 (1AS), Rg-B1 (1BS) and Rg-D1 (1DS)". In bread wheat, 3 dominant alleles Rg-A1b, Rg-B1b and Rg-D1b, are responsible for light-red, red and dark-red glume color respectively. In addition, black glume allele Rg-A1c and smokey-grey allele Rg-D1c exist. From these 5 alleles Rg-B1b is the most frequent [19-21, 26].

(2) Page 5 line 92. It is important to add here: "The biochemical nature of red, black and smokey-grey pigments in wheat glume remained unclear, however comparative analysis of the transcription of flavonoid biosynthesis structural genes in glumes of isogenic lines that differ by the allelic state of the Rg-A1 and Rg-D1 loci suggested Rg genes to be transcriptional activators in phlobaphene/3-deoxyanthocyanidin branch of flavonoid biosynthesis (article to be referred: DOI: 10.1134/S2079059713010085 (Khlestkina E.K. 2013. Genes Determining the Coloration of Different Organs in Wheat. Russ. J. Genet.: Appl. Res. 3:54-65.)."

(3) Page 13, Line 286: please replace "flavone 3-hydroxylase" by the correct enzyme name "flavanone 3-hydroxylase"

(4) Page 17, Lines 362-380: It is very strange that this part of discussion is focused on transposon insertion as mechanism for disruption of flavonoid biosynthesis only. Here it would be more appropriate to discuss first of all that the nearly 120 years old story of Rg genes studies in wheat successfully finished by identification of the Rg gene at the DNA level and discovering molecular mechanisms underlying this trait in wheat. It could be discussed also that it is the third branch of flavonoid pigments biosynthesis, for which the main regulatory gene was discovered (first was proanthocyanidin branch for red grain color [Himi, E. and Noda, K., Red Grain Colour Gene (R) of Wheat Is a MybType Transcription Factor, Euphytica, 2005, vol. 143, pp. 239-242.], second was anthocyanin branch for coloration of different parts of wheat plant [DOI: 10.1007/s00438-015-0991-0], while regulatory gene responsible for phlobaphene synthesis in red glume of wheat remained unsequenced... All these branches are governed by Myb-encoding transcriptional activators specific to each branch...

(5) Page 17, Line 364: please replace "flavan-4-ols" by correct name "flavan-4-ols"; furthermore change the phrase from simply "flavan-4-ols" to "such derivatives of flavan-4-ols as phlobaphenes and 3-deoxyanthocyanidins", because flavan-4-ols are non-colored, only their derivatives are colored substances in this branch of flavonoid biosynthesis pathway.

(6) Page 17, Lines 377-379: Authors say "Interestingly, we observed that light intensity had an effect on glume color in wheat which indicates that wheat glume color is also affected by environmental conditions" - please, re-write the end of the discussion, since this finding is too apparent and not new, it was noticed over 100 years ago and many times reported (summarized in Zeven 1983 (Zeven A.C. The character brown ear of bread wheat: a review // Euphytica 1983. V. 32. P. 299-310).

(7) Figure 3e: one group is named simply "flavonoid" - it should be specified, what group of flavonoid

substances is regulated by this group of Myb genes.

Reviewer #2 (Remarks to the Author):

In this nice manuscript the authors describe the diversity of a collection of spelt genotypes based on GBS, compare spelt and bread wheat diversity and provide an interesting hypothesis about spelt introduction in Europe. They then demonstrate how the availability of a pan-genome in wheat can help the dissection of a complex locus (26 gene paralogs in 10 Mb!) and the detection of structural variants and haplotypes associated to the red glume phenotype, a trait that segregates in both spelt and bread wheat germplasms. This is to my knowledge one of the first examples on how to exploit the wheat pan-genome and it may be of particular interest for the wheat community.

However, at this stage it really looks like there are two different and independent manuscripts, and I think the authors should try to better link these two sections. For example, is it possible to track the origin of the red glume phenotype by exploiting the collection assembled and the haplotype specific markers developed in this work? The title and the conclusions only refer to the identification of the MYB-TF associated to the red glume, that is of course the most interesting and novel part.

As for this second section, I have some major concerns:

- The association of group 3 alleles to the red glume phenotype is at the end based on the allele-specific marker, but I think the authors should try to give some information on the allelic variation at the molecular (e.g. protein) level. For example, is the premature stop codon (close to the CDS end according to Fig 3c) enough to say that rg1.hap7 is a pseudogene? This haplotype was also found in the two red genotypes. Similarly, group 4 alleles were ruled out because only found in white glume genotypes, but it is not clear how much they differ at the molecular level from the candidate group 3 alleles. Has the 47 bp InDel distinguishing group 3 alleles a possible effect on e.g. protein sequence/function, considering that it should be a gain of function mutation?
- The functional characterization of the dominant red allele Rg1.hap1 has to be compared with the recessive rg1.hap1 allele (or other recessive alleles). For example, the authors should measure the expression levels also for the MYB candidate gene in red and white spelt accessions, in addition to putative downstream genes involved in flavonoid biosynthesis. Is the MULE insertion affecting gene expression/function? Or protein sequence is different and may affect the regulation of downstream genes?
- Similarly, in the transient overexpression experiment, why the authors did not transform with the non-functional rg1.hap1 allele as well?
- The discussion (lines 357-374) should be improved. The trait may not be of direct breeding utility, but is there a biological meaning for glume pigmentation (e.g. a protection against high light intensity)? Any possible comparison with close species (e.g. barley with black or purple husk)? Is it possible to suggest the origin of the phenotype, also by screening more red spelt accessions with the allele specific markers developed in the manuscript? The red glume is dominant, and the diversification of paralog genes (gain of function mutation) is consistent with its inheritance. This might be somehow discussed in the manuscript.

As for the first part of the paper on spelt diversity, the tight molecular relatedness between Asian spelt and bread wheat is curious. Also, from fig 1a it seems that the spike of Asian spelt is more compact, like in wheat (they probably have the Q allele as suggested by the authors at line 319?). Are Asian spelts phenotypically well distinct from wheat, other than for brittle rachis and tenacious glumes? It may be that the accessions under examination come from bread wheat x spelt, then they were selected as "wheat-like" but with brittle rachis and tenacious glumes (only few genes/alleles introgressed from spelt?)

Other minor comments:

- Lines 419-420. Starting from 54,205 anchored SNPs, the authors have removed SNPs with rare alleles (MAF<5%) and high missing data (>20%), and that is ok. But, if LD= 1 means complete LD,

the authors have removed all the markers ($LD \leq 1$)? How many markers left at the end for the diversity analyses?

- GWAS results: how was the 2 Mb confidence interval defined? It was based on local LD?
- Gene paralogs: authors found extensive structural variations across a quite large genomic region (around 10 Mb), is it possible that the 26 paralogs are not allelic or that they have been mis-assembled/annotated at the Rg1 locus? Are there other highly similar genes (>93% similarity) on other chromosomes?
- Nomenclature of the functional alleles: does it make confusion to call the dominant alleles with the same numbers as the recessive ones? i.e. Rg1.hap2 is it numbered in this way because it is highly similar to rg1.hap2 and so on? Or it would be better to call them Rg1.hap11, Rg1.hap12 and Rg1.hap13?
- As for PI 190962, did the authors have working hypothesis on the possible second mutation they claim? Any possible evidence from the high-quality genomic sequence surrounding the Rg1 locus?

Reviewer #3 (Remarks to the Author):

The main objectives of the manuscript are to characterize a spelt collection by classically and advanced molecular technologies, to study the origin and history of spelta subspecies in Europe, and to identify and clone a transcription factor responsible of the red glume of spelta and other wheat accessions (Rg1). The manuscript is the logical continuation of a previous project '10+ Wheat Genomes Project' on the global variation of wheat genome (Walkowiak, S. et al. 2020 Spotlight on global wheat genome variation and impact for modern breeding. Nature under review).

A collection of 267 spelt accessions were genotyped by the genotyping-by-sequencing (GBS) technique, and the pertaining data analyses (principal component analysis, phylogeny, and admixture analysis) indicated three spelta gene pools comprising accessions collected in Asia, Central Europe, and the Iberian Peninsula. The origin and history of spelt were presented and discussed in detail providing evidence for two independent introductions of spelt into Europe.

The genome-wide association study (GWAS) with glume phenotypes extracted from historic data (Swiss National genebank) confirmed the location of the red glume gene (Rg1) on chromosome arm 1AS. The associated region contained 33 annotated genes in Chinese Spring, among which the authors identified a MYB transcription factor as candidate gene through the aptotype comparisons across ten wheat genomes. A haplotype specific marker based on a 47 bp InDel in the first intron was developed to validate the candidate Rg1.hap1 allele for the red glume. A RT-qPCR experiment in 20 spelt accessions using wheat flavonoid biosynthesis pathway genes and the transient expression of Rg1 in *Nicotiana benthamiana* indicated that the glume color at the Rg1 locus is controlled by a particular group of R2R3-MYB transcription factor variants.

Comments

The abstract briefly tells what was done and summarize the main results and conclusions. The author's contribution is placed in its proper perspective in relation to the state of knowledge. The subject is developed logically and effectively, and the manuscript is well organized and concise. The conclusions are adequate and supported by the data. The information is presented in a relatively simple and straightforward manner to be understood by a competent scientist or reader. Although the authors considered a morphological trait and a crop (spelta) of limited importance, the main merit of the manuscript is reporting as the high-quality genomes represents a valid and modern auxiliary for gene cloning in wheat. The subject falls within the scope of the Biology Communications.

1. The title of the manuscript might be modified to indicate not only the identification of the MYB

transcription factor regulating glume color but also the origin of spelta which is presented and discussed in detail.

2. The importance of the morphological marker "red glume" is too emphasized as currently molecular markers are widely used for the description of the variability of plant collections.

3. Line 47: 2. the scientific name of durum wheat (*Triticum durum*) should be reported as *Triticum turgidum* ssp. *durum* in order to standardize the wheat taxonomy in the text.

4. Standardize the references to the style of the Journal

Point-by-point responses (line numbers in our responses refer to the manuscript version with track changes)

Reviewer #1 (Remarks to the Author):

Association of hexaploid wheat genes, responsible for agronomically important traits, with certain genomic sequences remains a challenge even after whole genome sequencing of wheat. The results obtained in the current study can be estimated as valuable, because for the first time since the early 1900s when geneticists began to study red glume color trait in wheat, the Rg gene has been fully described at the DNA level. In addition, in the reviewed paper, some new insights to the origin and history of hexaploid spelt wheat are reported. The article can be accepted after following revisions:

(1) Page 5 line 90-92: It is more correct to say "Genetic studies revealed that glume color in hexaploid wheat is controlled by three loci on the homoeologous group 1 chromosomes: Rg-A1 (1AS), Rg-B1 (1BS) and Rg-D1 (1DS)". In bread wheat, 3 dominant alleles Rg-A1b, Rg-B1b and Rg-D1b, are responsible for light-red, red and dark-red glume color respectively. In addition, black glume allele Rg-A1c and smokey-grey allele Rg-D1c exist. From these 5 alleles Rg-B1b is the most frequent [19-21, 26].

>Our response: We have adapted the statement as suggested. The new statement reads as follows (lines 90-95): 'Genetic studies revealed that glume color in hexaploid bread wheat is controlled by the three loci Rg-A1 (1AS), Rg-B1 (1BS), and Rg-D1 (1DS) on the homoeologous group 1 chromosomes. Three dominant alleles Rg-A1b, Rg-B1b, and Rg-D1b are responsible for the red glume color, respectively, of which Rg-B1b (also referred to as Rg1) is the most frequent in hexaploid wheat. In addition, a black glume allele (Rg-A1c) and a smokey-grey glume allele (Rg-D1c) have been identified.'

(2) Page 5 line 92. It is important to add here: "The biochemical nature of red, black and smokey-grey pigments in wheat glume remained unclear, however comparative analysis of the transcription of flavonoid biosynthesis structural genes in glumes of isogenic lines that differ by the allelic state of the Rg-A1 and Rg-D1 loci suggested Rg genes to be transcriptional activators in phlobaphene/3-deoxyanthocyanidin branch of flavonoid biosynthesis (article to be referred: DOI: 10.1134/S2079059713010085 (Khlestkina E.K. 2013. Genes Determining the Coloration of Different Organs in Wheat. Russ. J. Genet.: Appl. Res. 3:54-65.))."

>Our response: We have added the statement as suggested (lines 98-101).

(3) Page 13, Line 286: please replace "flavone 3-hydroxylase" by the correct enzyme name "flavanone 3-hydroxylase"

>Our response: Typo was corrected.

(4) Page 17, Lines 362-380: It is very strange that this part of discussion is focused on transposon insertion as mechanism for disruption of flavonoid biosynthesis only. Here it would be more appropriate to discuss first of all that the nearly 120 years old story of Rg genes studies in wheat successfully finished by identification of the Rg gene at the DNA level and discovering molecular mechanisms underlying

this trait in wheat. It could be discussed also that it is the third branch of flavonoid pigments biosynthesis, for which the main regulatory gene was discovered (first was proanthocyanidin branch for red grain color [Himi, E. and Noda, K., Red Grain Colour Gene (R) of Wheat Is a MybType Transcription Factor, Euphytica, 2005, vol. 143, pp. 239-242.], second was anthocyanin branch for coloration of different parts of wheat plant [DOI: 10.1007/s00438-015-0991-0], while regulatory gene responsible for phlobaphene synthesis in red glume of wheat remained unsequenced... All these branches are governed by Myb-encoding transcriptional activators specific to each branch...

>Our response: We want to thank the reviewer for pointing out the historic relevance of the red glume phenotype for early genetic studies. We have added the following statement to put our work in perspective with these early genetic studies (lines 417-421): 'Because of its striking phenotype and importance for classifying bread wheat cultivars, the genetic inheritance of glume color was among the first traits that were systematically studied after the rediscovery of Mendel's laws in the early 1900s. Almost 120 years later we complement these pioneering studies with the identification of the DNA sequence that causes the red glume phenotype.'

We refrained, however, from entering into a deep discussion about the pigments involved in glume coloration. The aim of this study was to identify the genes conferring glume color through haplotype analysis. Although we have some evidence about the type of pigments involved, this needs to be followed up in future biochemical analyses.

(5) Page 17, Line 364: please replace "flavon-4-ols" by correct name "flavan-4-ols"; furthermore change the phrase from simply "flavan-4-ols" to "such derivatives of flavan-4-ols as phlobaphenes and 3-deoxyanthocyanidins", because flavan-4-ols are non-colored, only their derivatives are colored substances in this branch of flavonoid biosynthesis pathway.

>Our response: Thanks a lot for pointing this out. We have adapted the statement accordingly.

(6) Page 17, Lines 377-379: Authors say "Interestingly, we observed that light intensity had an effect on glume color in wheat which indicates that wheat glume color is also affected by environmental conditions" - please, re-write the end of the discussion, since this finding is too apparent and not new, it was noticed over 100 years ago and many times reported (summarized in Zeven 1983 (Zeven A.C. The character brown ear of bread wheat: a review // Euphytica 1983. V. 32. P. 299-310).

Our response: We modified our statement as follows (line 435-438): 'We observed that light intensity had an effect on glume color in wheat (Supplementary Fig. 13), which is in agreement with previous observations indicating that wheat glume color is affected by environmental conditions.'

(7) Figure 3e: one group is named simply "flavonoid" - it should be specified, what group of flavonoid substances is regulated by this group of Myb genes

Our response: Figure 3e has been updated accordingly.

Reviewer #2 (Remarks to the Author):

In this nice manuscript the authors describe the diversity of a collection of spelt genotypes based on GBS, compare spelt and bread wheat diversity and provide an interesting hypothesis about spelt introduction in Europe. They then demonstrate how the availability of a pan-genome in wheat can help the dissection of a complex locus (26 gene paralogs in 10 Mb!) and the detection of structural variants and haplotypes associated to the red glume phenotype, a trait that segregates in both spelt and bread wheat germplasms. This is to my knowledge one of the first examples on how to exploit the wheat pan-genome and it may be of particular interest for the wheat community.

However, at this stage it really looks like there are two different and independent manuscripts, and I think the authors should try to better link these two sections. For example, is it possible to track the origin of the red glume phenotype by exploiting the collection assembled and the haplotype specific markers developed in this work?

>Our response: We agree with the reviewer that the manuscript consists of two parts. However, we think that the population genomics and haplotype analyses are connected and complementary. Unfortunately, it is not possible to track the origin of the red glume phenotype. Glume color is an ancient trait that evolved in the ancestors of today's Triticum/Aegilops species. To highlight this fact we have included the following sentence in the introduction (lines 96-98): 'Homoeoalleles conferring dark glumes have also been described in various diploid wild wheat relatives of the genus Aegilops and in diploid and tetraploid Triticum species, indicating that genes controlling glume color were already present in the common ancestor of Triticeae'.

The title and the conclusions only refer to the identification of the MYB-TF associated to the red glume, that is of course the most interesting and novel part.

Our response: We have change the title as follows: 'Population genomics and haplotype analysis in spelt and bread wheat identifies a gene regulating glume color'.

As for this second section, I have some major concerns:

- The association of group 3 alleles to the red glume phenotype is at the end based on the allele-specific marker, but I think the authors should try to give some information on the allelic variation at the molecular (e.g. protein) level. For example, is the premature stop codon (close to the CDS end according to Fig 3c) enough to say that rg1.hap7 is a pseudogene? This haplotype was also found in the two red genotypes. Similarly, group 4 alleles were ruled out because only found in white glume genotypes, but it is not clear how much they differ at the molecular level from the candidate group 3 alleles. Has the 47 bp InDel distinguishing group 3 alleles a possible effect on e.g. protein sequence/function, considering that it should be a gain of function mutation?

Our response: In addition to the nucleotide sequence identity we now also indicate the amino acid identities in the new Supplementary Table 5. The premature stop codon in rg1.hap7 results in a putative

protein that lacks 66 amino acids at the C-terminus. This affects the regulatory domain and rg1.hap7 is thus most likely not functional. This is now indicated in line 214-216 as follows: 'rg-B1a_h7 had the transposon insertion in intron 2 and an additional premature stop codon in the last exon, resulting in a predicted protein that lacked 66 amino acids at the C-terminus compared to rg-B1a_h1'. As stated in line 241, the InDel marker was developed on an intronic sequence and the 47 bp InDel has thus no effect on the protein sequence.

- The functional characterization of the dominant red allele Rg1.hap1 has to be compared with the recessive rg1.hap1 allele (or other recessive alleles). For example, the authors should measure the expression levels also for the MYB candidate gene in red and white spelt accessions, in addition to putative downstream genes involved in flavonoid biosynthesis. Is the MULE insertion affecting gene expression/function? Or protein sequence is different and may affect the regulation of downstream genes?

Our response: We have included several new experiments to address this important point: First: In addition to the diagnostic group 3 specific marker, we developed new PCR markers specific for groups 1, 2, 4 and 5 based on group specific nucleotide polymorphisms. These markers were tested on the 53 Central European spelt accessions and the results were added to Supplementary table 6. This marker analysis revealed that group 1, 2, and 5 alleles do not correlate with glume color and these alleles were thus excluded as Rg-B1 candidates. Interestingly, all red glume spelt accessions were positive for the group 4 alleles in addition to the group 3 alleles. We thus also tested the group 4 specific markers on the Japanese bread wheat diversity panel (Supplementary Table 7). Eight white glume wheat accessions were positive for the group 4 alleles, but lacked the group 3 alleles. All red accessions were positive for the group 3 and 4 alleles. This could indicate that although the group 4 alleles are not determining glume color, they might be required in addition to a group 3 allele for the red glumes. We also carefully re-analyzed the ten wheat genome sequences and found that the red glume cultivars Jagger and Norin 61 carry a group 4 allele with 100% sequence identity to the rg-B1a-h9 allele of Mace on chromosomes unanchored and 1D, respectively (new Supplementary Fig. 9). In summary, the new marker analysis shows that the group 3 alleles determine glume color, but that presence of a group 4 allele might be necessary for the red glume phenotype in addition to a group 3 allele. However, presence of a group 4 allele alone results in white glumes. These results are described in lines 238-280. Second: We measured expression of the various haplotypes. We were able to show expression of group 3 and group 4 alleles in spelt accessions with red glumes. Group 1, 2, and 5 alleles were not expressed. This is described in the main text as follows (lines 284-286): 'The group 3 and 4 alleles were expressed in glumes of red glume spelt accessions, while expression of the group 1, 2, and 5 alleles was not detectable in red and white glume accessions.'

- Similarly, in the transient overexpression experiment, why the authors did not transform with the non-functional rg1.hap1 allele as well?

Our response: We also infiltrated group 1, 4, and 5 alleles into Nicotiana. The results are summarized in Supplementary Fig. 12d. We show that group 4 and 5 alleles also induced a fluorescent peak similar to the one found after infiltrating group 3 alleles. This indicates that several Rg-B1 alleles have the potential to induce flavonoid biosynthesis when overexpressed in Nicotiana. Group 1 alleles did not induce flavonoid biosynthesis, which is possibly caused by a deletion of one amino acid in the highly conserved R2 motif.

These results are described in the main text as follows (lines 354-360): ‘Infiltration of a group 4 (rg-B1a_h9) and a group 5 (rg-B1a_h10) allele also resulted in a fluorescent peak, while the group 1 allele rg-B1a_h2 did not (Supplementary Fig. 12d). These results indicate that group 4 and 5 alleles also have the potential to induce flavonoid production when overexpressed in an artificial system. The group 1 alleles rg-B1a_h2, rg-B1a_h3 and rg-B1a_h4 carry a deletion of a conserved amino acid in the N-terminal R2 MYB domain, which might explain why they were not functional in Nicotiana.’

- The discussion (lines 357-374) should be improved. The trait may not be of direct breeding utility, but is there a biological meaning for glume pigmentation (e.g. a protection against high light intensity)? Any possible comparison with close species (e.g. barley with black or purple husk)? Is it possible to suggest the origin of the phenotype, also by screening more red spelt accessions with the allele specific markers developed in the manuscript? The red glume is dominant, and the diversification of paralog genes (gain of function mutation) is consistent with its inheritance. This might be somehow discussed in the manuscript.

Our response: As indicated by reviewer 1 we have added a historic perspective of the red glume phenotype in the discussion (lines 417-421). We are not aware of any biological advantage of red glumes. Also, it is unfortunately not possible to deduce the origin of the red glume phenotype since homoeoallelic genes controlling dark glumes have also been described in various Aegilops species as well as diploid and tetraploid wheats.

As for the first part of the paper on spelt diversity, the tight molecular relatedness between Asian spelt and bread wheat is curious. Also, from fig 1a it seems that the spike of Asian spelt is more compact, like in wheat (they probably have the Q allele as suggested by the authors at line 319?). Are Asian spelt phenotypically well distinct from wheat, other than for brittle rachis and tenacious glumes? It may be that the accessions under examination come from bread wheat x spelt, then they were selected as “wheat-like” but with brittle rachis and tenacious glumes (only few genes/alleles introgressed from spelt?)

Our response: This is indeed a very interesting point. The close genetic relatedness of Asian spelt and bread wheat has been reported previously (see for example Blatter et al. (2004) TAG 108:360-367). To our knowledge, there is no clear and unambiguous definition of spelt other than a hexaploid wheat (AABBDD genome) with a speltoid (elongated) spike, brittle rachis and tenacious glumes. Phenotypically, all Asian spelt accessions in our collection have a brittle rachis and tenacious glumes comparable to European spelt. But as correctly indicated by the reviewer, this difference between Asian spelt and bread wheat might be caused by few genes (the Q gene or other genes). We discuss this point in lines 384-392.

Other minor comments:

- Lines 419-420. Starting from 54,205 anchored SNPs, the authors have removed SNPs with rare alleles (MAF<5%) and high missing data (>20%), and that is ok. But, if LD= 1 means complete LD, the authors have removed all the markers (LD≤1)? How many markers left at the end for the diversity analyses?

Our response: This was a mistake. We have set LD=1 to not discard SNPs based on linkage disequilibrium (the default parameter in SNPhylo is 0.1). We only filtered SNPs based on MAF and missing data to construct the phylogenetic tree, which resulted in 23,554 SNPs. This is now corrected.

- GWAS results: how was the 2 Mb confidence interval defined? It was based on local LD?

Our response: This was defined by LD, i.e. the location of the SNPs that did not show association.

- Gene paralogs: authors found extensive structural variations across a quite large genomic region (around 10 Mb), is it possible that the 26 paralogs are not allelic or that they have been mis-assembled/annotated at the Rg1 locus? Are there other highly similar genes (>93% similarity) on other chromosomes?

Our response: We have added a phylogenetic tree showing that these alleles are indeed true paralogs that form a distinct cluster (new Supplementary Fig. 8). We have carefully and manually re-annotated all the alleles in the intervals and are convinced about the accuracy of the annotation. The group 3 and group 4 alleles were validated by sequencing of the cDNA. Mis-assemblies can never be ruled out completely. However, it needs to be stated that these ten wheat genomes represent the best genomic resources in wheat (probably in all cereals) that are currently available.

- Nomenclature of the functional alleles: does it make confusion to call the dominant alleles with the same numbers as the recessive ones? i.e. Rg1.hap2 is it numbered in this way because it is highly similar to rg1.hap2 and so on? Or it would be better to call them Rg1.hap11, Rg1.hap12 and Rg1.hap13?

Our response: The nomenclature of the alleles was slightly adapted to be in line with the suggested 'rules for gene nomenclature in wheat: 2020' that were discussed in the wheat community recently. In particular, we know make a distinction between functional (Rg-B1b) and non-functional alleles (rg-B1a) as it is already defined in the wheat gene catalogue. According to the document mentioned above, haplotypes are numbered starting from 1 for each allele.

- As for PI 190962, did the authors have working hypothesis on the possible second mutation they claim? Any possible evidence from the high-quality genomic sequence surrounding the Rg1 locus?

Our response: It is very likely that the possible second-site mutation is unlinked and not in the Rg1 locus. An unlinked suppressor gene of red glumes has already been described and we refer to this work in the main text as follows (lines 310-311): 'An unlinked suppressor of glume color in wheat has been described on chromosome 3A'.

Reviewer #3 (Remarks to the Author):

The main objectives of the manuscript are to characterize a spelt collection by classically and advanced molecular technologies, to study the origin and history of spelta subspecies in Europe, and to identify and

clone a transcription factor responsible of the red glume of spelta and other wheat accessions (Rg1). The manuscript is the logical continuation of a previous project ‘10+ Wheat Genomes Project’ on the global variation of wheat genome (Walkowiak, S. et al. 2020 Spotlight on global wheat genome variation and impact for modern breeding. Nature under review).

A collection of 267 spelt accessions were genotyped by the genotyping-by-sequencing (GBS) technique, and the pertaining data analyses (principal component analysis, phylogeny, and admixture analysis) indicated three spelta gene pools comprising accessions collected in Asia, Central Europe, and the Iberian Peninsula. The origin and history of spelt were presented and discussed in detail providing evidence for two independent introductions of spelt into Europe.

The genome-wide association study (GWAS) with glume phenotypes extracted from historic data (Swiss National genebank) confirmed the location of the red glume gene (Rg1) on chromosome arm 1AS. The associated region contained 33 annotated genes in Chinese Spring, among which the authors identified a MYB transcription factor as candidate gene through the aplotype comparisons across ten wheat genomes. A haplotype specific marker based on a 47 bp InDel in the first intron was developed to validate the candidate Rg1.hap1 allele for the red glume. A RT-qPCR experiment in 20 spelt accessions using wheat flavonoid biosynthesis pathway genes and the transient expression of Rg1 in *Nicotiana benthamiana* indicated that the glume color at the Rg1 locus is controlled by a particular group of R2R3-MYB transcription factor variants.

Comments

The abstract briefly tells what was done and summarize the main results and conclusions. The author’s contribution is placed in its proper perspective in relation to the state of knowledge. The subject is developed logically and effectively, and the manuscript is well organized and concise. The conclusions are adequate and supported by the data. The information is presented in a relatively simple and straightforward manner to be understood by a competent scientist or reader. Although the authors considered a morphological trait and a crop (spelta) of limited importance, the main merit of the manuscript is reporting as the high-quality genomes represents a valid and modern auxiliary for gene cloning in wheat. The subject falls within the scope of the *Biology Communications*.

1. The title of the manuscript might be modified to indicate not only the identification of the MYB transcription factor regulating glume color but also the origin of spelta which is presented and discussed in detail.

Our response: We gave the manuscript the following new title: ‘Population genomics and haplotype analysis in spelt and bread wheat identifies a gene regulating glume color’

2. The importance of the morphological marker “red glume” is too emphasized as currently molecular markers are widely used for the description of the variability of plant collections.

Our response: We agree with the reviewer that molecular markers are becoming increasingly important in describing wheat cultivars and plant collections. However, it is important to mention that ‘glume color’ is still an important trait that is frequently used to describe, differentiate and characterize wheat cultivars. For example, glume color is still mentioned in the *Plant Varieties Journals* of many countries upon the

acceptance and release of new wheat cultivars (for examples, see references 35 and 36 or <https://www.inspection.gc.ca/english/plaveg/pbrpov/journal/grants117e.shtml#whe>). This is what we refer to by mentioning glume color as an 'important' trait.

3. Line 47: 2. the scientific name of durum wheat (*Triticum durum*) should be reported as *Triticum turgidum* ssp. *durum* in order to standardize the wheat taxonomy in the text.

Our response: done

4. Standardize the references to the style of the Journal

Our response: done

REVIEWERS' COMMENTS:

Reviewer #2 (Remarks to the Author):

In this revised manuscript, authors have complemented their work by adding new results and text, that overall answer to the points I have raised in my review. Only, I cannot see any data (or figure) supporting the statement added at lines 284-286): 'The group 3 and 4 alleles were expressed in glumes of red glume spelt accessions, while expression of the group 1, 2, and 5 alleles was not detectable in red and white glume accessions.'

I think the manuscript can now be accepted for publication on Communications Biology.

In this revised manuscript, authors have complemented their work by adding new results and text, that overall answer to the points I have raised in my review. Only, I cannot see any data (or figure) supporting the statement added at lines 284-286): ‘The group 3 and 4 alleles were expressed in glumes of red glume spelt accessions, while expression of the group 1, 2, and 5 alleles was not detectable in red and white glume accessions.’

Our response: We have addressed the one remaining reviewer comment by providing details about our gene expression study as follows (lines 273-275): ‘Sequencing of 97 cDNA clones from red glume spelt accessions representing all haplotype groups revealed that the group 3 and 4 alleles were expressed, while expression of the group 1, 2, and 5 alleles was not detectable in red and white glume accessions.’